# Lindqvist versus Keggin-Type Polyoxometalates as Catalysts for Effective Desulfurization of Fuels

Simone Fernandes [ID], Fátima Mirante, Baltazar de Castro, Carlos M. Granadeiro *[ID] and Salete S. Balula *[ID]

REQUIMTE/LAQV & Department of Chemistry and Biochemistry, Faculty of Sciences, University of Porto, 4169-007 Porto, Portugal; up201603496@edu.fc.up.pt (S.F.); fatimamirante@hotmail.com (F.M.); bcastro@fc.up.pt (B.d.C.)
* Correspondence: cgranadeiro@fc.up.pt (C.M.G.); sbalula@fc.up.pt (S.S.B.)

**Abstract:** A correlation between polyoxotungstate structures and their catalytic performance for oxidative desulfurization processes was investigated. Bridged lanthanopolyoxometalates that incorporate identical metallic centers with Keggin- $Eu[PW_{11}O_{39}]^{11-}$ and Lindqvist-type $[Eu(W_5O_{18})_2]^{9-}$ structures were used as catalysts for the oxidation of the most representative refractory sulfur compounds. Both compounds were able to desulfurize a multicomponent model diesel under sustainable conditions, i.e., using ionic liquid as an extraction solvent and hydrogen peroxide as an oxidant. However, the Lindqvist catalyst appeared to achieve complete desulfurization faster than the Keggin catalyst while using a lesser amount of catalyst and oxidant. Furthermore, the reusable capacity of the Lindqvist-type $[Eu(W_5O_{18})_2]^{9-}$ was confirmed for consecutive oxidative desulfurization processes. The contribution of the lanthanide metallic center for the catalytic performance of these compounds was investigated by studying the analogous $[TB(W_5O_{18})_2]^{9-}$ compound. Identical desulfurization efficiency was obtained, even reusing this catalyst in consecutive reaction cycles. These results indicate that the active catalytic center of these compounds is probably related to the octahedral tungsten centers. However, a higher number of tungsten centers in the polyoxometalate structure did not result in higher catalytic activity.

**Keywords:** polyoxometalates; Keggin-type; Lindqvist-type; desulfurization; oxidative catalysis; ionic liquid; hydrogen peroxide



## 1. Introduction

Fossil fuels are now, and will be for the next few decades, the major source of energy, especially for maritime, road and air transportation. Therefore, the elimination emissions from sulfur-derived products during fuel combustion continues to be a crucial topic for investigation. This includes the application of sustainable and efficient desulfurization processes, capable of producing fuels that have strict policies for sulfur contents in fuels: ultra-low for road transportation (<10 ppm), and more recently, a global limit of sulfur in fuel ships of 0.5% (m/m) [1], set by the International Marine Organization. Hydrodesulfurization (HDS) is the traditional method used by petroleum industry, which requires severe operational conditions and is less efficient in removing the aromatic sulfur compounds present in fuels [2,3]. Different complementary desulfurization processes have been studied based on extraction, adsorption, oxidation or even biological methods [1,4,5]. The oxidative desulfurization method (ODS) is an alternative process that allows for a highly efficient removal of refractory sulfur compounds under mild and eco-sustainable conditions [1,6–8]. Furthermore, the ODS process allows for the treatment of more viscous and less volatile fuels derived from the heavy fuel oils. The ODS method involves three steps: a selective oxidant to convert sulfides into corresponding sulfones or sulfoxides, followed by the extraction of sulfones, and the recovery of the catalyst [9,10].

Polyoxometalates (POMs) are clusters of anionic metal-oxides that are thermal and oxidative stable compounds. The chemical properties of POMs such as redox potential,

electron-transfer properties, acidities and solubility, can be finely changed by adjusting the metal ions, the heteroatoms and the counter-cations [11]. These properties make POMs superior catalytic materials for oxidative catalysis. Our research group has been investigating novel POM-based eco-sustainable oxidative desulfurization systems for the production of sulfur-free fuels [7,12–15]. Most of these studies have been performed with a derived Keggin-type structure $[PM_{12}O_{40}]^{3-}$, with M = $Mo^{IV}$, $W^{IV}$ or $V^{V}$. Some of these studies used lanthanopolyoxometalates (LnPOMs) obtained through the coordination of lanthanide ions, such as $Eu^{3+}$, $Tb^{3+}$, $Sm^{3+}$, etc. to coordinative POM lacunary fragments, namely $[PM_{11}O_{39}]^{7-}$ units with M = W and Mo [16]. In general, LnPOMs exhibit interesting luminescent properties and other specific characteristics, which result from the synergy between the properties of the lanthanide ions and POM units. The LnPOMs have a large number of applications: optics, catalysis, electronics, magnetics, biomedicine and luminescence [17–23]. The Keggin-type ($[XM_{12}O_{40}]^{n-}$) anions are mainly studied in catalysis and also in the oxidative desulfurization processes. This is not only due to the high catalytic efficiency of these types of POMs, but also to their properties and catalytic activity, which can be modulated by the substitution of the M addenda atoms or the heteroatom X by different metal centers. Moreover, several Keggin-type derivates can be formed by removing one or more $MO^{4+}$ units, resulting in lacunary compounds which contain free oxygen atoms, that can readily coordinate with transition metals or even lanthanide ions. Sandwich-type lanthanopolyoxotungstates $[Ln(PW_{11}O_{39})_2]^{11-}$ ($Ln^{3+}$ = $Eu^{3+}$, $Tb^{3+}$) are efficient catalysts for ODS [11,12,15,24]. The Lindqvist-type $[Ln(W_5O_{18})_2]^{9-}$ ($Ln^{3+}$ = $Eu^{3+}$, $Tb^{3+}$, $Gd^{3+}$, $La^{3+}$) has also been used to catalyze the oxidation of sulfur compounds in fuels [11,25–27]. The Lindqvist-type is an iso-polyoxometalate with only one type of transition metal atom in its structure and multiwavelength systems that allow the excitation wavelength to be tuned by variation of the lanthanide center or through the coordination of an organic ligand to the POM.

This work presents for the first time a comparative study using Keggin- and Lindqvist-type LnPOMs as catalysts for desulfurization technology. Furthermore, the influence of the number of tungsten centers in the POM structure and the nature of the lanthanide is evaluated here for the first time. The potassium salts of the sandwich monovacant Keggin- and Lindqvist-type phosphotungstates were tested in the desulfurization of the model diesel containing three different sulfur compounds with a total sulfur concentration of 1500 ppm. The reusability and stability of the catalysts have also been investigated.

## 2. Results

### 2.1. Catalysts Characterization

The $Eu^{3+}$ based POMs, Lindqvist-type $[Eu(W_5O_{18})_2]^{9-}$ and Keggin-type $[Eu(PW_{11}O_{39})_2]^{9-}$, are clearly the most studied POMs, and already described in the literature by different techniques, such as elemental and thermal analysis, powder X-ray diffraction, vibrational (FTIR and FT-Raman), $^{31}P$ NMR and photoluminescence spectroscopy, to prove the authenticity and purity of the compounds [11,28–30].

Spectroscopic methods including FTIR and $^{31}P$ NMR were used to characterize the europium compounds. The infrared spectrum of $[Eu(W_5O_{18})_2]^{9-}$ presents several bands in the range 700–1200 $cm^{-1}$, the terminal $\nu_{as}(W=O_t)$ stretch ca.931 $cm^{-1}$ and the $\nu_{as}(W-O-W)$ corner or edge-shared stretching modes between 700–900 $cm^{-1}$ (Figure S1 in Supplementary Materials). The infrared spectrum of $Eu(PW_{11})_2$ displays four characteristic strong asymmetric vibration bands for the Keggin-type frameworks: $\nu_{as}(P-O)$ between 1100–1040 $cm^{-1}$, terminal $\nu_{as}(W-O_t)$ at ca. 950 $cm^{-1}$, corner-sharing $\nu_{as}(W-O_b-W)$ at ca. 850 $cm^{-1}$, and edge-sharing $\nu_{as}(W-O_c-W)$ at ca. 800 $cm^{-1}$. $^{31}P$ NMR spectroscopy was also used to identify and characterize the potassium salt of $Eu(PW_{11})_2$ structure in $D_2O$ solution, showing as expected, a singlet at 0.36 ppm [28]. These results are in accordance with the literature data and indicate that they were successfully prepared. The thermal behaviour of the lanthanopolyoxotungstates was investigated by thermogravimetric analysis (TGA) and the obtained TGA curves are exhibited in Figure S2 in Supplementary Materials. The

Lindqvist-type POMs exhibit a main weight loss until 150 °C that can be assigned to the to the evaporation of physisorbed water (hydration water molecules). An additional weight loss can be observed in the TGA curve of $Tb(W_5O_{18})_2$ in the 150–325 °C range, most likely due to the presence of chemisorbed water. The TGA curve of Keggin-type POM shows a longer initial weight loss step until ca. 200 °C (hydration water molecules) followed by another weight loss in the 350–550 °C range, which suggests the degradation of the sandwich-type structure.

## 2.2. Desulfurization Studies

The preliminary studies with potassium salts of Keggin- $(Eu(PW_{11})_2)$ and Lindqvist-type $(Eu(W_5)_2)$ POMs were performed using a model diesel containing three refractory sulfur compounds present in real diesel that are the most difficult to desulfurize: dibenzothiophene (DBT), 4-methyldibenzothiophene (4-MDBT) and 4,6-dimethyldibenzothiophene (4,6-DMDBT), with an individual concentration of approximately 500 ppm of sulfur containing compounds in *n*-octane. These oxidative desulfurization studies were performed using a biphasic system (1:1 model diesel/[BMIM]PF$_6$, extraction solvent). The process procedures in two main steps: first an initial liquid-liquid extraction in the presence of the catalyst by stirring for 10 min at 70 °C (initial extraction); and then, after the initial extraction equilibrium is reached, the catalytic oxidation stage is initiated by adding the hydrogen peroxide oxidant (promoting the oxidation of sulfur compounds to the corresponding sulfoxides and/or sulfones in the extraction solvent). The oxidation occurs in the extraction phase, since no oxidation products were detected in the model diesel phase. However, when the sulfur compounds are catalytically oxidized, more sulfur compounds from the diesel phase can be transferred to the extraction phase. Therefore, during the catalytic oxidation stage, a continuous extraction of sulfur occurs (ECODS, extraction catalytic oxidative desulfurization system). The comparison of the catalytic performance of the homogeneous Lindqvist and the Keggin catalysts for the ECODS was performed, by varying the catalyst and oxidant amounts. The initial conditions adopted were 0.75 mL of model diesel, 0.75 mL of extraction solvent [BMIM]PF$_6$ and 75 μL of $H_2O_2$ at 70 °C. These were optimized conditions previously reported by the research group for similar catalytic systems [31]. Ionic liquids have been demonstrated to have an effective effect as extractive desulfurization solvents. Between these, the [BMIM]PF$_6$ have demonstrated to be most efficient [8,14,16]. The effect of the catalyst amount was studied using 0.3 and 3 μmol of catalyst and the results obtained are displayed in Figure 1. The desulfurization profile using Linqvist-type catalyst $Eu(W_5)_2$ is less influenced by the amount of catalyst, and higher catalytic efficiency is achieved using the lower amount of catalyst. Using this catalyst, total desulfurization was achieved after only 1 h of reaction. Slightly higher catalytic efficiency was found using 3 μmol than 0.3 μmol only during the first minutes. The desulfurization profile of the model diesel when catalysed by the Keggin-type catalyst $Eu(PW_{11})_2$ demonstrates that complete desulfurization was faster achieved using the highest amount of catalyst. Therefore, 3 μmol of Keggin-type catalyst were needed to achieve complete desulfurization after 2 h (instead of 79% obtained using 0.3 μmol of $Eu(PW_{11})_2$). However, after 3 h, complete desulfurization was achieved using 3 and 0.3 μmol of $Eu(PW_{11})_2$ catalyst.

The amount of oxidant presents in the ECODS system is another important parameter that can have a high influence in the catalyst performance. The $H_2O_2$ oxidant is activated by its interaction with the catalyst, forming active species. These are usually hydroperoxy- or peroxo-POM species that are able to oxidize the sulfur compounds into the corresponding sulfoxides through a nucleophilic attack. The initial POM is further regenerated. The subsequent oxidation of the sulfoxides leads to the formation of sulfones. The mechanism involved in ECODS system using polyoxotungstate catalysts is well reported in the literature [12,24,32–35]. In the oxidant amount study, 75 and 100 μL of $H_2O_2$ were used. Results obtained using Keggin and Linqvist catalysts are displayed in Figure 2. Using the Lindqvist catalyst, the increase in oxidant amount decreased the desulfurization efficiency of the ECODS system. This is probably caused by the occurrence of some deactivation of the cata-

lyst or even by the difficulty of sulfur extraction in the presence of a higher amount of water content in the system. Visually, the aqueous oxidant is located between the apolar model diesel phase and the polar ionic liquid [BMIM]PF$_6$ extraction phase. In both Keggin and Lindqvist catalytic systems, it is possible to observe a decrease in the initial extraction data when the oxidant amount is increased, i.e., the extractive desulfurization achieved after the first 10 min. However, using the Keggin catalyst, it is possible to observe an increment in the desulfurization efficiency during the oxidative catalytic step. Therefore, the Keggin catalyst also needs a higher amount of oxidant to achieve the same desulfurization efficiency as the Lindqvist catalyst obtained after 1 h, which corresponds to complete desulfurization of model diesel. When 75 μL of H$_2$O$_2$ was used instead of 100 μL, only 85% of desulfurization was obtained after 1 h. From the catalyst and oxidant amounts study is possible to verify that the Lindqvist catalyst achieved complete desulfurization faster, using less amount of catalyst and less amount of oxidant. This demonstrates that the Lindqvist (Eu(W$_5$)$_2$) compound is a more efficient catalyst than the Keggin (Eu(PW$_{11}$)$_2$). Therefore, the number of atomic tungsten centers in the POM structure does not have a direct correlation with the activity of the compound. A higher dimensional polyoxotungstate structure containing a higher number of tungsten centers may not be more catalytically active for an oxidative reaction as others with a smaller size and lower number of tungsten centers. Moreover, the primary phosphorus atomic center in the Keggin (Eu(PW$_{11}$)$_2$) structure does not have an active participation in catalyst activity.

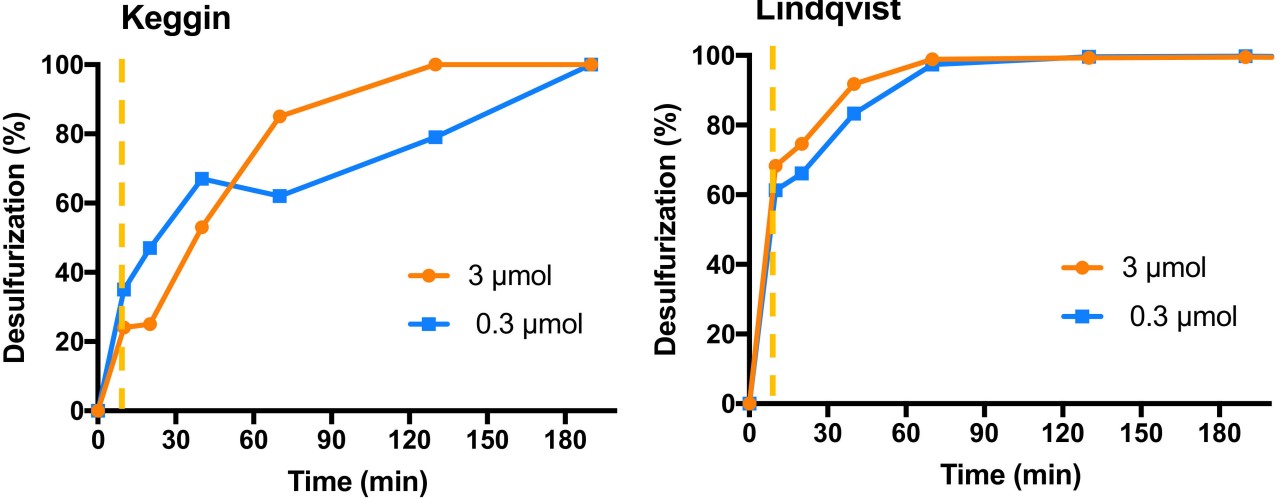

**Figure 1.** Desulfurization profile using 0.3 and 3 μmol of Keggin-type [Eu(PW$_{11}$O$_{39}$)$_2$]$^9$ (**right**) and Lindqvist-type [Eu(W$_5$O$_{18}$)$_2$]$^{9-}$ (**left**) catalysts to treat the multicomponent model diesel in a biphasic diesel/[BMIM]PF$_6$ (1:1) ECODS system, using and 75 μL of H$_2$O$_2$ at 70 °C. The vertical dashed line indicates the time that the oxidative catalytic reaction was started by the addition of the oxidant.

### 2.2.1. Reusing Homogeneous POMs

The reusability capacity of both the structures, Keggin and Lindqvist POMs, was investigated under the ECODS system, using 3 μmol of catalyst and 75 μL of H$_2$O$_2$. The reusability of both catalysts was evaluated for five consecutive cycles. At the end of each ECODS cycle, the desulfurized model diesel was removed from the system and a new sulfurized model diesel and oxidant sample were added to perform a consecutive ECODS cycle, maintaining the same experimental conditions and the POM/[BMIM]PF$_6$ catalytic active phase. Figure 3 displays the results of desulfurization obtained for five ECODS consecutive cycles, after 1 h of reaction. The Lindqvist catalyst maintained its catalytic performance during the 5 consecutive cycles, whereas the Keggin one slightly increased its activity after the first cycle that was maintained for over 4 consecutive ECODS cycles. This behavior indicates that some catalytic activation must occur using the Keggin POM. In fact, the oxidant needs to be activated by the interaction with the catalyst, forming peroxo

compounds as catalytic active intermediates [16]. In the case of the Keggin POM catalyst, a slower interaction with the $H_2O_2$ oxidant must have occurred, when compared to the Lindqvist POM which caused the increased catalytic activity observed.

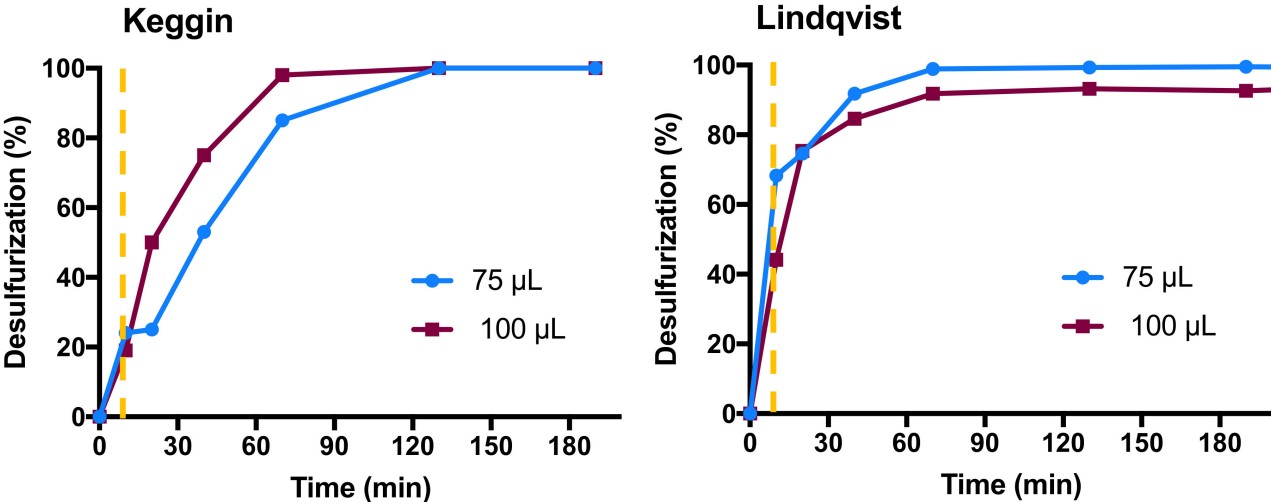

**Figure 2.** Desulfurization profile of the multicomponent model diesel in a biphasic diesel/[BMIM]PF$_6$ (1:1) ECODS system, using 3 μmol of catalyst and 100 and 75 μL of $H_2O_2$ at 70 °C. The vertical dashed line indicates the time that the oxidative catalytic reaction was started by the addition of the oxidant.

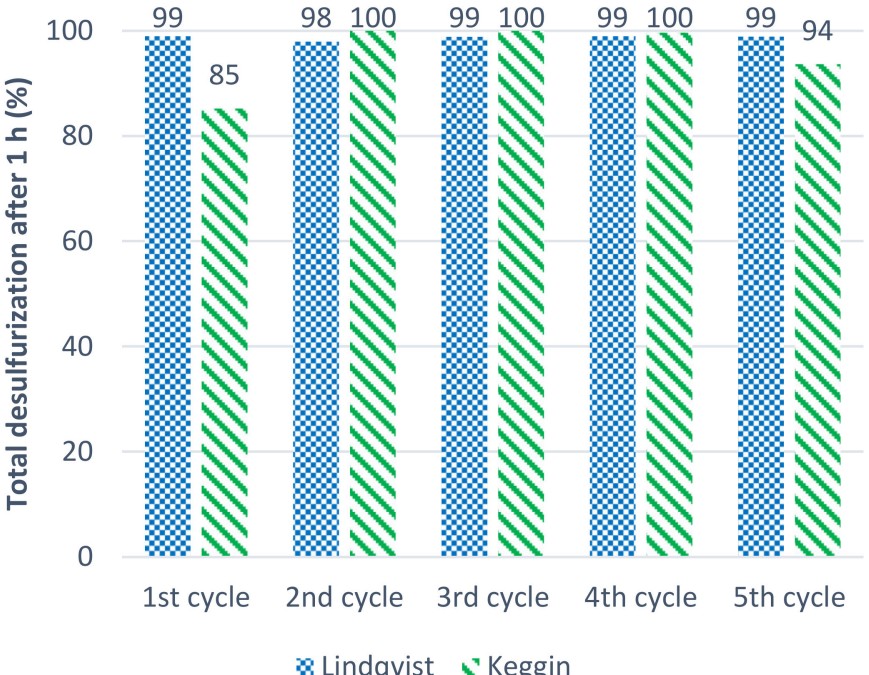

**Figure 3.** Desulfurization results obtained after 1 h of 5 consecutive ECODS diesel/[BMIM]PF$_6$ processes, maintaining the same POM/[BMIM]PF$_6$ catalytic phase, using the Keggin and the Lindqvist catalysts (3 μmol) and $H_2O_2$ (75 μL) at 70 °C.

### 2.2.2. Influence of Lanthanide Nature

To investigate the contribution of the lanthanide as a catalytic active center, the catalytic activity of the europium Lindqvist POM Eu(W$_5$)$_2$ was compared with the analogous terbium Tb(W$_5$)$_2$ compound (Figure 4). Similar desulfurization profile of the multicomponent model diesel was obtained between both Lindqvist compounds, which indicates that the bridge lanthanide metallic center should not have an important contribution on the

catalytic performance of the POM (Figure 4). Furthermore, $Tb(W_5)_2$ was reused for several ECODS cycles and also for this Lindqvist POM, and its activity was maintained for at least five consecutive ECODS cycles (Figure 5).

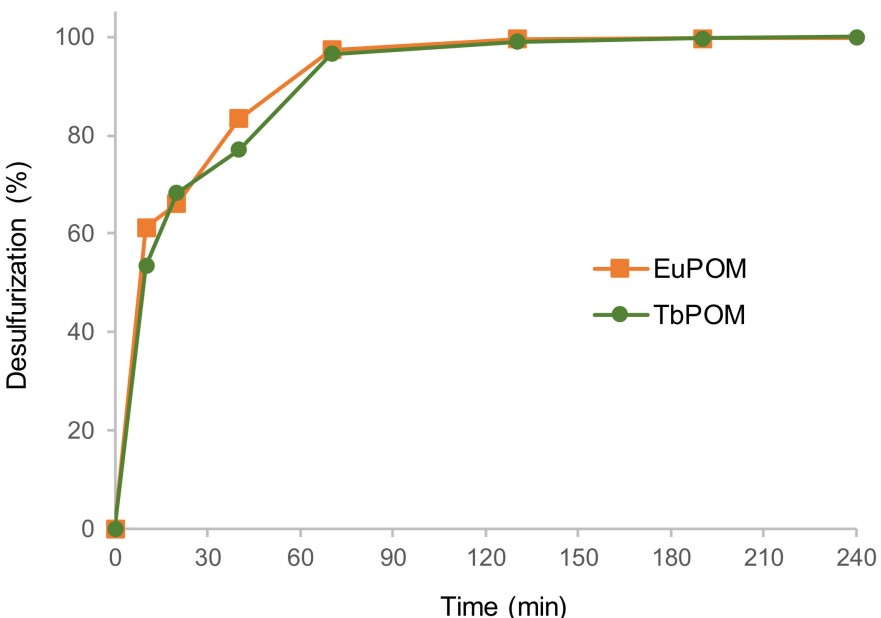

**Figure 4.** Desulfurization profile of the multicomponent model diesel in a biphasic diesel/[BMIM]PF$_6$ (1:1) ECODS system, using 3 μmol of Lindqvist catalyst ($Eu(W_5)_2$ or $Tb(W_5)_2$) and 75 μL of $H_2O_2$ at 70 °C.

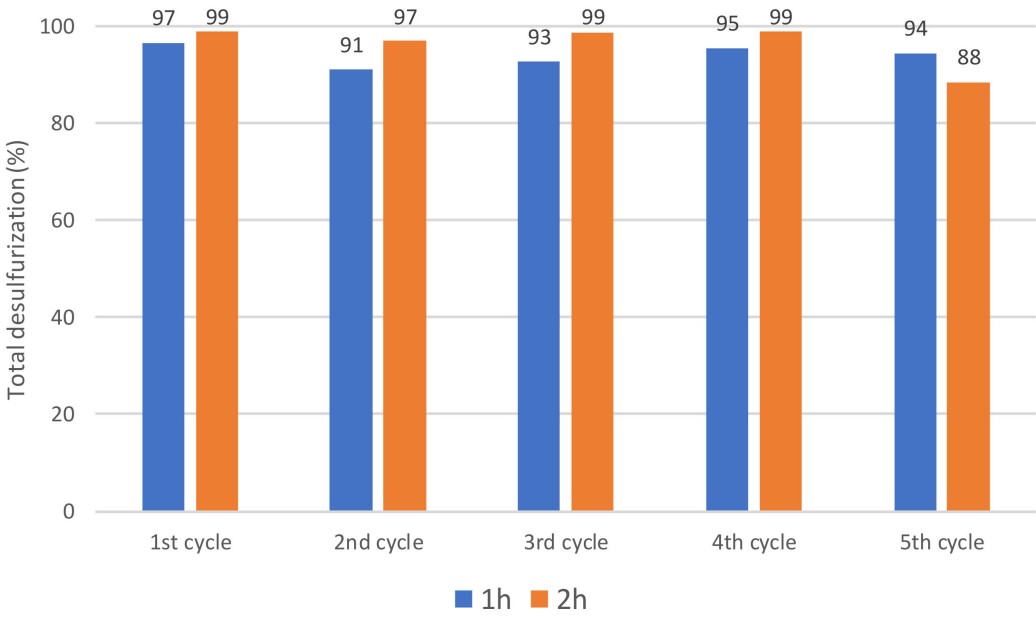

**Figure 5.** Desulfurization results obtained after 1 h and 2 h of five consecutive ECODS diesel/[BMIM]PF$_6$ cycles, maintaining the same POM/[BMIM]PF$_6$ catalytic phase, using $Tb(W_5)_2$ Lindqvist catalysts (3 μmol) and $H_2O_2$ (75 μL) at 70 °C.

### 2.2.3. Comparison between Different POM Structures

A comparison of the desulfurization efficiency to treat model diesel catalyzed by different POM structures in the presence of $H_2O_2$ as oxidant is presented in Table 1. It is possible to observe that the structure of the POM catalysts play an important role in the desulfurization efficiency; however, the extraction solvent also presents an important

influence. In general, ionic liquids appeared to promote a higher desulfurization efficiency than the acetonitrile. Different studies using the Keggin-type $Eu(PW_{11})_2$ catalyst were performed and faster complete desulfurization was achieved in this study using $[BMIM]PF_6$ solvent [15,24]. Furthermore, the lanthano-bridged POM Keggin-type ($Eu(PW_{11})_2$) presents a higher structural stability than the pristine Keggin $[PM_{12}O_{40}]^{3-}$ (M = W and Mo) and the monovacant $[PW_{11}O_{39}]^{7-}$, when used as catalysts for oxidative desulfurization of identical multicomponent model diesel [7,12,31]. Only a few examples could be found in the literature using the lanthano-bridged POM with the Lindqvist structure, and these only used single-model diesel containing only DBT, which is more easily oxidized than the BT or the DBT derivatives [25,27]. In this study, identical results were obtained using sulfur multicomponent model diesel.

**Table 1.** Extractive/Oxidative Desulfurization efficiency of model diesel, using $H_2O_2$ as oxidant and various POM structures as homogeneous catalysts.

| Catalyst | Solvent | Time (h) | Desulfurization (%) | Ref. |
|---|---|---|---|---|
| $[PMo_{12}O_{40}]^{3-}$ | $[BMIM]PF_6$ | 2 | 100 [a] | [34] |
| $[PW_{12}O_{40}]^{3-}$ | $[BMIM]PF_6$ | 1 | 100 [a] | [12] |
| $[PW_{11}O_{39}]^{7-}$ | No | 2 | 96.5 [a] | [7] |
| $[PW_{11}Zn(H_2O)O_{39}]^{5-}$ | $CH_3CN$ | 4 | 100 | [10] |
| $[Eu(PW_{11}O_{39})_2]^{11-}$ | $CH_3CN$ | 2 | 73.9 | [15] |
| $[Eu(PW_{11}O_{39})_2]^{11-}$ | $CH_3CN$ | 4 | 100 | [24] |
| $[Eu(PW_{11}O_{39})_2]^{11-}$ | $[BMIM]PF_6$ | 1 | 100 | This work |
| $[Sm(Pmo_{11}O_{39})_2]^{11-}$ | $[BMIM]PF_6$ | 1.5 | 100 | [24] |
| $[EuW_{10}O_{36}]^{9-}$ | $[omim]PF_6$ | 0.5 | 100 [b] | [25] |
| $[LaW_{10}O_{36}]^{9-}$ | $[omim]PF_6$ | 0.5 | 100 [b] | [27] |
| $[EuW_{10}O_{36}]^{9-}$ | $[BMIM]PF_6$ | 1 | 100 | This work |
| $[TbW_{10}O_{36}]^{9-}$ | $[BMIM]PF_6$ | 1 | 100 | This work |

[a] Low structural stability of the POM catalyst was found. [b] Only DBT (instead of a multicomponent sulfur model diesel) was used to study the desulfurization efficiency.

## 2.2.4. Stability of Keggin- and Lindqvist-Type Polyoxometalates

The stability of the Europium Keggin- and Lindqvist-type POMs were evaluated by comparison of the emission spectra, before and after catalysis. The studied EuPOMs exhibit peculiar photoluminescent properties, with an efficient energy transfer process to the lanthanide emitting center, as a result of the coordinated POM moieties [11,36]. For this reason, we prepared aqueous solutions, 1 mM of the as-prepared Lindqvist and Keggin POMs, and studied its emission under excitation into the maximum of the W-O charge transfer band (285 nm). The results confirm the occurrence of the characteristic intramolecular energy transfer to the $Ln^{3+}$, with both emission spectra displaying the typical $^5D_0 \rightarrow {}^7F_J$ (J = 2–4) emission peaks of $Eu^{3+}$ (Figure 6) [28,37]. The EuPOM/$[BMIM]PF_6$ phases were recovered after catalytic use and its emission spectra were acquired under the same experimental conditions for comparison purposes. As expected, the spectra of the recovered phases show peaks with considerably lower emission intensity when compared with the as-prepared samples due to the higher dispersion of the EuPOMs in the $[BMIM]PF_6$. Nevertheless, both spectra are composed by the same $Eu^{3+}$ emission peaks assigned to the $^5D_0 \rightarrow {}^7F_J$ (J = 2–4) transitions (Figure 6—insets). Moreover, the relative intensities of the $^5D_0 \rightarrow {}^7F_J$ transitions are still preserved in the emission spectra of the recovered phases, which could indicate that the Keggin and Lindqvist POM structures are retained after catalytic use since these transitions are known to be extremely sensitive to changes in the local symmetry of $Eu^{3+}$ [29,38,39].

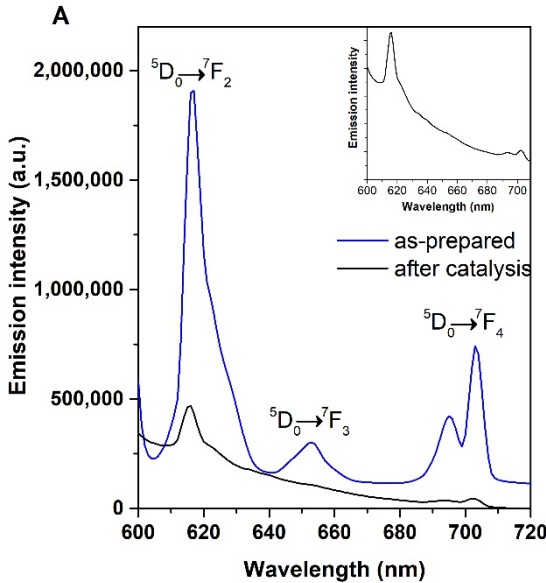
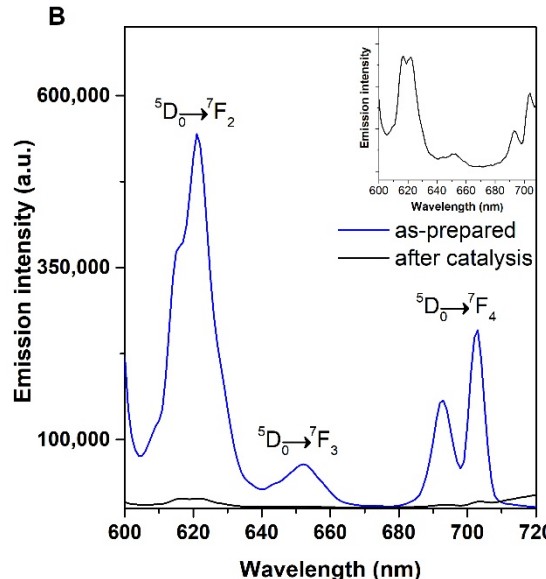

**Figure 6.** Emission spectra of (**A**) Keggin-type and (**B**) Lindqvist-type POMs as-prepared (blue) and the LnPOM/[BMIM]PF$_6$ phases after catalytic use (black) with an excitation wavelength of 285 nm.

## 3. Experimental Section

### 3.1. Materials and Methods

All the reagents used in the preparation of the polyoxometalates, namely sodium tungstate dihydrate (Aldrich, St. Louis, MO, USA, 99%), glacial acetic acid (Merck, Kenilworth, NJ, USA), potassium chloride (Merck), europium chloride hexahydrate (Sigma, St. Louis, MO, USA, 99.9 %,), terbium (III) chloride hexahydrate (Aldrich, St. Louis, MO, USA, 99.90%), hydrochloric acid (Panreac, Barcelona, Spain, 37 %) were used as received without further purification. The reagents for catalytic studies, including dibenzothiophene (DBT, Aldrich, St. Louis, MO, USA), 4-methyldibenzothiophene (4-MDBT, Aldrich, St. Louis, MO, USA), 4,6-dimethyldibenzothiophene (4,6-DMDBT, Alfa Aeser, Haverhill, MA, USA), *n*-octane (Aldrich), 1-butyl-3-methylimidazolium hexafluorophosphate ([BMIM]PF$_6$, Sigma-Aldrich) and 30% *w/w* hydrogen peroxide (Aldrich, St. Louis, MO, USA) were purchased from commercial sources. Infrared absorption spectra were recorded for 400–4000 cm$^{-1}$ regions on a Jasco 460 Plus spectrometer (Jasco, Tokyo, Japan), using KBr pellets. Thermogravimetric analysis was performed using a Hitachi STA 7200 RV equipment (Hitachi, Tokyo, Japan) under inert atmosphere (nitrogen flow of 200 mL/min), at room temperature up to 500 °C with a heating rate of 5 °C/min. $^{31}$P NMR spectra for liquid solutions were recorded with a Bruker Avance III 400 spectrometer (Bruker, Billerica, MA, USA), and chemical shifts are given with respect to external 85% H$_3$PO$_4$. Emission spectra were acquired in a Horiba Fluorolog-QM fluorometer (Horiba, Kyoto, Japan) with an excitation wavelength of 285 nm. Catalytic reactions were periodically monitored by GC-FID analysis carried out in a Bruker 430-GC-FID chromatograph (Bruker, Freemont, CA, USA). Hydrogen was used as carrier gas (55 cm s$^{-1}$) and fused silica Supelco capillary columns SPB-5 (30 m × 0.25 mm i. d.; 25 µm film thickness) were used.

### 3.2. Synthesis of Lanthanopolyoxometalates

#### 3.2.1. Lindqvist-Type POM

The potassium salts of the [Ln(W$_5$O$_{18}$)$_2$]$^{9-}$-type [Ln (III) = Eu and Tb] lanthanopolyoxometalates were prepared using an adaptation of the Weakley et al. method [36,40]. Briefly, an aqueous solution of sodium tungstate dihydrate (15.2 mmol, 7 mL) was prepared and the pH adjusted to 7 by addition of glacial CH$_3$COOH. The solution was heated at 90 °C and a hot aqueous solution of LnCl$_3$·6H$_2$O (1.52 mmol; 2 mL) was added dropwise followed by an aqueous solution of KCl (17.5 mmol; 8 mL). The mixture was allowed to stir

at 90 °C for 30 min and stored in the refrigerator for 3 days. The obtained precipitate was filtered, washed with ethanol, and dried in a desiccator over silica gel. Thermogravimetric analyses were performed for the determination of the hydration water molecules.

$K_9[Eu(W_5O_{18})_2]\cdot7H_2O$ (abbreviated as Eu(W$_5$)$_2$). TGA showed a mass loss of 4.2% up to 150 °C (loss of seven $H_2O$ hydration molecules). Selected FT-IR (cm$^{-1}$): 931, 833, 791, 708.

$K_9[Tb(W_5O_{18})_2]\cdot5H_2O$ (abbreviated as Tb(W$_5$)$_2$). TGA showed a mass loss of 3.3% up to 150 °C (loss of five $H_2O$ hydration molecules). Selected FT-IR (cm$^{-1}$): 933, 843, 793, 706.

### 3.2.2. Keggin-Type POM

The potassium salt of europium, $K_{11}[Eu(PW_{11}O_{39})_2]\cdot5H_2O$ (abbreviated as Eu(PW$_{11}$)$_2$) was prepared by following a modified literature procedure [28]. A solution of EuCl$_3$·6H$_2$O was added dropwise to an aqueous solution of the precursor ligand ($K_7[PW_{11}O_{39}]\cdot10H_2O$; abbreviated as PW$_{11}$), previously prepared [41]. The mixture was stirred for 1 h at 90 °C. PW$_{11}$ and Eu$^{3+}$ were dissolved in the minimum volume of water, and both solutions were added in rigorously stoichiometric amounts to prepare Eu(PW$_{11}$)$_2$ (1:2). Elemental analysis: calcd (%). Eu 2.5, K 7.1, P 1.1, W 66.9; found (%) Eu 2.9, K 7.5, P 1.8, W 67.7. Selected FT-IR (cm$^{-1}$): 1096, 1040, 942, 878, 798, 698. TGA showed a mass loss of 1.4% in the range 50–150 °C (loss of five $H_2O$ hydration molecules).

### 3.3. Oxidative Desulfurization Studies

The ODS studies were performed using a model diesel containing three sulfur compounds: dibenzothiophene (DBT), 4-methyldibenzothiophene (4-MDBT) and 4,6-dimethyldibenzothiophene (4,6-DMDBT), in *n*-octane (with a total sulfur concentration of 1500 ppm). The experiments were carried out under air in a closed borosilicate 5 mL vessel, equipped with a magnetic stirrer and immersed in a thermostatically controlled liquid paraffin bath at 70 °C. In a typical experiment, 3 μmol of POM was added to 1:1 model diesel/extraction solvent (750 μL each). The ionic liquid, 1-butyl-3-methylimidazolium hexafluorophosphate ([BMIM]PF$_6$), was used as extraction solvent. The resulting mixture was stirred for 10 min, after which the catalytic step was initiated by adding 30% *w/v* hydrogen peroxide (75 μL, H$_2$O$_2$/S = 14). The sulfur content in the model diesel was periodically quantified by GC analysis using tetradecane as the external standard. The reusability of the catalyst was evaluated by removing the desulfurized model diesel at the end of each ODS cycle and adding a new portion of model diesel and oxidant, maintaining the condition.

## 4. Conclusions

The catalytic performance of Europium-bridged polyoxometalates with Keggin- Eu[PW$_{11}$O$_{39}$]$^{11-}$ and Lindqvist-type [Eu(W$_5$O$_{18}$)$_2$]$^{9-}$ structures was investigated for the oxidative desulfurization of a multicomponent model diesel. The Lindqvist catalyst showed a slightly higher catalytic performance than the Keggin type, since complete desulfurization was achieved after only 1 h using less amount of oxidant. Furthermore, the catalytic activity of the Lindqvist compounds was investigated using the analogous [Tb(W$_5$O$_{18}$)$_2$]$^{9-}$ structure. In this case, identical catalytic performance was obtained and this result suggests that the Lanthanide center does not contribute to the catalytic efficiency of the Lindqvist compounds. The reusability of the catalysts was confirmed for at least five desulfurization cycles and the stability of the Keggin and the Lindqvist structures were confirmed by their photoluminescent properties.

**Supplementary Materials:** The following supporting information can be downloaded at: https://www.mdpi.com/article/10.3390/catal12060581/s1, Figure S1: FT-IR spectra of Eu(W5)2, Tb(W5)2 and Eu(PW11)2; Figure S2: TGA curves for (A) Eu(W5)2, (B) Tb(W5)2 and (C) Eu(PW11)2.

**Author Contributions:** Conceptualization, S.S.B. and C.M.G.; methodology, S.F. and F.M.; investigation, S.F. and F.M.; writing—original draft preparation, S.F. and S.S.B.; writing—review and editing,

C.M.G.; supervision, B.d.C. and S.S.B.; funding acquisition, S.S.B. All authors have read and agreed to the published version of the manuscript.

**Funding:** This work was supported by the Fundação para a Ciência e Tecnologia (FCT), Portugal, by the strategic project UIDB/50006/2020 for LAQV. S.S.B. thanks FCT/MCTES for funding through the Individual Call to Scientific Employment Stimulus Ref. CEECIND/03877/2018. C.M.G acknowledges FCT/MCTES for funding through DL 57/2016 program contract –Norma transitória.

**Conflicts of Interest:** The authors declare no conflict of interest.

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
