# Peer review of "Lindqvist versus Keggin-Type Polyoxometalates as Catalysts for Effective Desulfurization of Fuels"

_catalysts, doi:10.3390/catal12060581_

Round 1
Reviewer 1 Report
Please see the attachment.

Reviewer 2 Report
In the work Keggin Eu[PW11O39]11- and Lindqvist-type [Eu(W5O18)2] structures were used as catalysts for the oxidation of the sulfur compounds (DBT, 4-MDBT and 4,6-DMDBT) in model diesel. Authors investigated the influence of a number of parameters on the course of the model reaction. I would like to recommend the manuscript for publication after minor revisions:
On what basis was the ionic liquid selected? Have other ionic liquids been tested?
Other comments:
Figure 1: change the description in the legend of Figure 1.
line 193 and 198 no full stop at the end of the sentence.
Line 221: change the word “ative”
Author Response
Review 2
In the work Keggin Eu[PW11O39]11- and Lindqvist-type [Eu(W5O18)2] structures were used as catalysts for the oxidation of the sulfur compounds (DBT, 4-MDBT and 4,6-DMDBT) in model diesel. Authors investigated the influence of a number of parameters on the course of the model reaction. I would like to recommend the manuscript for publication after minor revisions:
On what basis was the ionic liquid selected? Have other ionic liquids been tested?
Responses from Authors:
The authors acknowledge the reviewer for the important comments. The modifications introduced in the manuscript following reviewer suggestions were highlighted at green.
In the section 3.2, the importance of using BMIMPF6 ionic liquid was introduced (lines from 177). In fact, Ionic liquids were selected as extraction solvents, as an alternative to conventional organic solvents, due to their high thermal stability, non-volatility and recycle capacity. This information was supported by references
Other comments:
Figure 1: change the description in the legend of Figure 1.
Response: This was modified and it was highlighted at green.
line 193 and 198 no full stop at the end of the sentence.
Response: This was modified and it was highlighted at green.
Line 221: change the word “ative”
Response: This was modified and it was highlighted at green.

Reviewer 3 Report
The authors presented an interesting work demonstrating applications of polyoxometalates as catalysts for effective desulfurization. The whole manuscript is clearly written, and there are minimum confusing graph labels and other content. All the minor comments which could improve the already good manuscript are listed below. To sum up, this work fully fits the journal scope. Therefore I recommend the manuscript to be published in Catalysts after minor revisions.
Minor comments:
line 51 - correct the superscript in the vanadium oxidation state
line 174, 175, and 182 - decimal separator --> usage of a comma instead of the dot.
Figure 1 - label for the yellow dotted line is not in English. What the adding H2O2 mean? Is the slove in desulfurization different before and after the adding step? Could the author discuss the slopes as a sign of the desulfurization rate? (At least for me, the slope looks almost identical both before and after the H2O2 addition.)
line 198 - punctuation (full stop) mark is missing.
Figure 2 - use the regular sign for the greek letter mu in the legend
line 233 - typo in the word Lindqvist
Figure 3 - the catalyst stability is excellent. However, an interesting increase in activity after the first catalytic run is presented for Keggin type. Could the authors discuss these in more detail? The statement "This behaviour indicates that some catalytic activation must occurs using the Keggin POM. " is vague.
Author Response
Review 3
The authors presented an interesting work demonstrating applications of polyoxometalates as catalysts for effective desulfurization. The whole manuscript is clearly written, and there are minimum confusing graph labels and other content. All the minor comments which could improve the already good manuscript are listed below. To sum up, this work fully fits the journal scope. Therefore I recommend the manuscript to be published in Catalysts after minor revisions.
Minor comments:
line 51 - correct the superscript in the vanadium oxidation state
line 174, 175, and 182 - decimal separator --> usage of a comma instead of the dot.
Figure 1 - label for the yellow dotted line is not in English. What the adding H2O2 mean? Is the slove in desulfurization different before and after the adding step? Could the author discuss the slopes as a sign of the desulfurization rate? (At least for me, the slope looks almost identical both before and after the H2O2 addition.)
line 198 - punctuation (full stop) mark is missing.
Figure 2 - use the regular sign for the greek letter mu in the legend
line 233 - typo in the word Lindqvist
Figure 3 - the catalyst stability is excellent. However, an interesting increase in activity after the first catalytic run is presented for Keggin type. Could the authors discuss these in more detail? The statement "This behaviour indicates that some catalytic activation must occurs using the Keggin POM. " is vague.
Responses from Authors:
The authors acknowledge the reviewer for the important comments. All the comments were accepted and corrected as suggested. The modifications introduced in the manuscript following reviewer suggestions were highlighted at grey.
The label for the yellow dotted corrected and it was added a new sentence in Figure 1 and 2 “The vertical dashed line indicates the instant that oxidative catalytic reaction was started by addition of the oxidant”.
In the text line 162 was mentioned the two main steps of oxidative desulfurization process: first an initial liquid-liquid extraction by stirring at 70 ºC (initial extraction); after 10 minutes the equilibrium is reached, the catalytic oxidation stage is initiated by adding the hydrogen peroxide oxidant (promoting the oxidation of sulfur compounds to the corresponding sulfoxides and/or sulfones in the extraction solvent medium).
About the increasing activity of Keggin POM after the first ECODS cycle, may be caused by a slower interaction of the Keggin catalyst with the H2O2 oxidant, when compared to the Lindqvist POM. The catalyst need to interact with the oxidant for its activation. In this step, peroxocompounds can be formed as catalytic active intermediates. This information was supported by reference [16].

Reviewer 4 Report
Comments are attached.

Author Response
Review 4
This manuscript deals on the study of some lanthano polyoxometalates that incorporate identical metallic centers with Keggin Eu[PW11O39]11- and Lindqvist-type [Eu(W5O18)2]9− structures which were used as catalysts for the oxidation of the most representative refractory sulfur compounds.
The manuscript contains valuable information of interest for the wide scientific community interested in this kind of compounds. In my opinion the manuscript could be considered for publication in present form with only some observations:
- The authors claim that: “Thermogravimetric analysis were performed from room temperature up to 800 ˚C … ”, but they present only some results up to 150 ˚C. My recommendation is to give more details (tables, graphs) about thermal behaviour of these compounds up to 800 ˚C.
Responses from Authors: The authors acknowledge the comment of the referee. The thermogravimetric analysis were performed from room temperature up to 550 ˚C (there was a typo in the Experimental section). The TGA curves of the lanthanopolyoxometalates were added to the Supporting Information and their thermal behaviour was discussed (highlighted in blue).
My recommendation is to add more data and eventually some graphs of IR analysis.
Responses from Authors: The authors acknowledge the comment of the referee. For that purpose, the IR graphs of the lanthanopolyoxometalates were introduced in the Supporting Information along with the band assignment.
Page 5: Grammar mistake in the phrase: “…structure do not have an ative participation in catalyst activity”
Authors Response: The authors have corrected the mentioned phrase in the manuscript (highlighted in blue).
References: [31] was written with capital letters.
Authors Response: The authors have changed the capital letters in reference 31.

Round 2
Reviewer 1 Report
Accept in present form